# Lymphadenopathies before and during the Pandemic COVID-19: Increasing Incidence of Metastases from Solid Tumors

**DOI:** 10.3390/jcm11236979

**Published:** 2022-11-26

**Authors:** Stefania Trasarti, Raffaele Troiano, Mario Biglietto, Silvia Sorella, Chiara Lisi, Giovanni Manfredi Assanto, Luisa Bizzoni, Gregorio Antonio Brunetti, Carla Giordano, Emma Rullo, Mariangela Saracino, Paolina Saullo, Marco Vignetti, Maurizio Martelli, Roberto Caronna

**Affiliations:** 1Department of Translational and Precision Medicine, Sapienza University of Rome, 00185 Rome, Italy; 2Department of Surgical Science, Sapienza University of Rome, 00161 Rome, Italy; 3Department of Radiological Sciences, Oncology and Pathology, Sapienza University of Rome, 00161 Rome, Italy

**Keywords:** lymphadenopathies, pandemic COVID-19, hematological malignancies, metastatic cancers, solid tumors, lymph-node biopsy, screening

## Abstract

Since December 2019, the world has experienced a pandemic caused by SARS-CoV-2, a virus which spread throughout the world. Anti-COVID19 measures were applied to limit the spread of the infection, affecting normal clinical practice. In 2020, studies on the possible impact of the pandemic considering the screening programs for early diagnosis of cancer were conducted, resulting in a prediction of delayed diagnosis of cancer. We performed a retrospective monocentric study on patients who present with the onset of lymphadenomegalies evaluated at our Hematological Department from February 2019 to October 2021 and undergoing excisional lymph-node biopsy. Three periods were considered: pre-pandemic, first pandemic period and second pandemic period (Group A, B and C). We included 258 patients who underwent a surgical biopsy and received a histological diagnosis. Hematological evaluation of outpatients sent by the general practitioner and surgical biopsies did not decrease among the three groups, despite limitations placed during this pandemic as well as new diagnoses of hematological malignancies. However, the diagnosis of metastatic cancer significantly increased from 2019 (7.8%) to 2021 (22.1%) (*p* = 0.042). Our data supports the hypothesis that the pandemic affected the national screening programs of early cancer detection.

## 1. Introduction

Since March 2020, to limit the spread of the COVID 19 pandemic, national governments’ approaches have included two major measures: health policies aimed to enhance the capacity of the healthcare system and policies aimed to control the viral diffusion, such as lockdowns and social distancing [1,2]. This has resulted in a positive impact on the pandemic, with a reduction in the COVID-19 cases in many countries [3], and to many deaths being prevented by the expansion of intensive care units. On the downside, this has led to an increased difficulty for patients with symptoms related to oncological diseases to access diagnostic procedures, with a subsequent delay in the diagnosis and the start of treatment. Clinicians were therefore faced with advanced malignant diseases [4]. 

The study was driven during 2021 by our finding of two patients with metastatic lymph nodes that were not usually involved by the primary tumor: one right supraclavicular nodal metastasis from prostatic cancer and one right cervical nodal metastasis from ovarian cancer. Both were included in the study. The aim of this research was to evaluate a possible increase in patients with lymphadenomegaly related to cancer metastasis or hematological malignancies during the COVID-19 pandemic.

## 2. Materials and Methods

We performed an observational retrospective monocentric study, including patients affected by lymphadenomegaly who came for the first time, at the Hematological Department of Sapienza University/ Policlinico Umberto I of Rome between May 2019 and October 2021. All patients underwent excisional lymph-node biopsy by the surgical team. 

Clinical and histopathological data were collected through the revision of clinical files. This study respects the principles of the declaration of Helsinki. All patients were listed into a database and divided according to period of diagnosis: Group A (pre-pandemic period), from 1 May 2019 to 29 February 2020; Group B (first pandemic period), from 1 March 2020 to 31 December 2020; Group C (second pandemic period), from 1 January 2021 to 31 October 2021.

For each patient, we considered age, sex, the date of access to the Hematological Department, excisional lymph-node biopsy date and histological result. Patients with a previous history of cancer or hematological malignancy, patients who did not undergo excisional lymph-node biopsy and those lost after the first clinical evaluation were excluded from this study. 

## 3. Statistical Analysis

Descriptive statistics, such as the frequency (n), arithmetic mean, and standard deviation (SD), are presented for normally distributed variables. Otherwise, medians and the standard errors (SE) with interquartile ranges (25 and 75 percentile) are used. To compare differences between the groups, the χ^2^-test was used for categorical variables and the Mann–Whitney *U*-test for continuous variables. A *p* value of 0.05 was considered significantly. Statistical analysis was performed with SPSS statistics.

## 4. Results

We included 258 patients with lymphadenopathy who underwent excisional biopsy in three different periods. One hundred and two patients composed Group A (pre-pandemic period): they were part (2.7%) of 3765 visits for hematological diseases performed between May 2019 and February 2020. Forty-five were male and 57 were female, and the average age was 58.6 years. In this group, the excisional biopsy pathological exam revealed 70 hematological malignancies (68.6%), 9 metastatic cancers (8.8%), and 23 non-neoplastic diseases (22.5%). Eighty-eight patients composed Group B (first pandemic period): they were part (4.43%) of 1984 visits (4.43%) for hematological diseases examined between March 2020 and December 2020. Forty-two were male and 46 were female, and the mean age was 60 years. In this group, the excisional biopsy pathological exam revealed 48 hematological malignancies (54.5%), 17 metastatic cancers (19.3%), and 23 non-neoplastic diseases (26.1%). Sixty-eight patients composed Group C (second pandemic period): they were part (2.7%) of 2458 visits for hematological diseases examined between January 2021 and October 2021. Thirty-six were male and 32 were female, and the mean age was 52 years. In this group, the excisional biopsy pathological exam revealed 36 hematological malignancies (52.9%), 15 metastatic cancers (22%), and 17 non-neoplastic diseases (25%) (Table 1). 

It is worth noting that some patients had unusual localizations of lymph-node metastasis distant from the site of the primary solid tumor: 6/17 cases in the first pandemic period (group B) and 8/15 cases in the second pandemic period (Group C).

Univariate analysis showed a significative difference in terms of the incidence of metastatic cancer from Group A (7.8%) to Group C (22.8%) (*p* = 0.042, Figure 1) (Table 1). Univariate analysis was also performed for sex, age, and number of evaluated patients. No significative differences were evidenced between the three groups.

## 5. Discussion

Lymphoadenomegaly frequently occurs and may be the first sign of a malignancy. It is mandatory for an accurate diagnosis to proceed with a physical examination, blood tests, imaging and, if pathological ultrasound features are present, an excisional lymph-node biopsy. 

Lymph-node specimen collection followed by histological evaluation plays a key-role, either in defining the lymphoma subtype or in diagnosing other pathologies which can show up with a swollen lymph-node, such as inflammatory diseases or solid cancers. In the latter, the presence of a distal lymph-node metastasis defines an advanced tumor stage.

In this context, timing in diagnosis is crucial and can influence the prognosis, either for lymphoma or for other types of cancer [4].

The management of the coronavirus infection has led to the discontinuation of clinical follow-ups, therapies, or surgery for patients with chronic diseases [4]. At the same time, healthy subjects also did not undergo screening tests for the early diagnosis of cancer. Moreover, in the case of symptoms or signs suggestive of malignancies, these patients had many difficulties accessing diagnostic paths and turned to the hospital when diseases were already in an advanced stage [5,6].

In the first phase of the pandemic (2020), studies on the possible impact of the pandemic on the screening programs for early diagnosis of cancer were conducted, resulting in a prediction of delayed diagnosis of cancer and stressing the importance of continuing screening programs [7,8]. Our findings are consistent with the international literature regarding the diagnostic delay.

Subsequent studies have confirmed the predicted diagnostic delay [9,10,11,12,13,14,15,16,17,18,19]. In a study conducted in Taiwan, Tsai et al. [9] reported that admissions to breast outpatient clinics in hospitals decreased by 37% during the lockdown period, and breast cancer screening decreased by 22%. In the UK, compared to the first 6 months of 2019, the diagnosis of breast cancer decreased by 16% in the first 6 months of 2020 [10]. Koca et al. [11] found that the number of admissions to breast outpatient clinics decreased by 26.3% and the number of screening mammographs decreased by 79.8% throughout Turkey in 1 year during the pandemic; furthermore, tumor size, axillary lymph node positivity, and neoadjuvant chemotherapy were higher during the pandemic (*p* =   0.005, *p*  =  0.012, *p =* 0.042, respectively). 

We observed an expected reduction in the number of first hematological examinations in 2020 (Group B, first pandemic period), but there was no significant difference between the number of excisional lymph-node biopsies performed in the pre-pandemic period and in the first pandemic one. We have hypothesized, therefore, that fewer patients requested a hematological consultation or fewer patients were sent to the hematologist, while the hospital was able to maintain adequate activity despite the pandemic.

Therefore, we assume that the increasing number of patients with lymphadenomegaly related to metastases from solid tumors is only partially the consequence of a lower number of patients visiting hospital, but is mainly the result of a difficulty accessing diagnostic procedures, with a further increase of cases in the second pandemic period (2021). 

It is also interesting to note that two patients in group C with metastatic solid cancer were diagnosed with distant lymph nodes not usually involved (one right supraclavicular nodal metastasis from prostatic cancer and one right cervical nodal metastasis from ovarian can.

Given the limitations of a retrospective study, with a small cohort of patients, we reported this finding: we observed a consistent number of lymphoadenomegalies involved with metastatic solid cancers and often not contiguous to the primary site, evidence of a late diagnosis.

It has been reported that SARS-CoV-2 alters the immune system by interfering with mitochondrial function. Given the role of the immune system in the development and spread of tumors, it has been hypothesized that COVID-19 could promote tumor growth and/or metastatic spreading [20,21,22]. We cannot confirm this hypothesis because we have not evaluated the incidence of COVID infection. Further studies are necessary and expected in this context.

## 6. Conclusions

During the COVID-19 pandemic, the management of neoplastic diseases became more difficult, especially for solid tumors. Despite the national restrictions, the clinical activity in our center was never interrupted, and at times the tool of telemedicine was used to secure this. In this study, we investigated a side of the healthcare system which could have been and was affected by the pandemic. Our data were consistent with the international literature regarding the impact of diagnostic delay, stressing the importance of early screening programs.

## Figures and Tables

**Figure 1 jcm-11-06979-f001:**
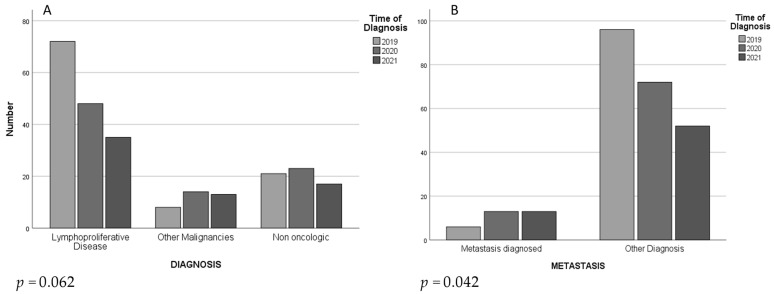
This figure shows number per year of diagnosis performed through lymph-node biopsy. There is a decreasing trend of lymphoproliferative disease (**A**) and a significant increasing of distal metastases (**B**) diagnosed from 2019 to 2021.

**Table 1 jcm-11-06979-t001:** Patients’ data. This table shows cohort characteristics, highlighting sex, site of biopsy, diagnosis, and metastasis. Univariate analysis was performed with an χ^2^-test, and *p* values are reported on the right side.

Patients		Time of Diagnosis								
		2019 102 pts		2020 88 pts		2021 68 pts		Total		*p* Value
		N.	%	N.	%	N.	%	N.	%	
Sex	Male	45	44.1	42	47.7	36	52.9	123	47.7	0.052
	Female	57	55.9	46	54.1	32	47.1	135	52.3	
Site of Biopsy	Neck	57	55.9	47	55.3	43	66.2	147	58.3	0.41
	Axilla	20	19.6	17	20.0	11	16.9	48	19.0	
	Inguinal	20	19.6	12	14.1	6	9.2	38	15.1	
	Mediastinal/Abdomen	5	4.9	8	9.4	5	7.7	18	7.1	
Diagnosis	Lymphoproliferative Disease	70	68.6	48	54.5	36	52.9	154	59.7	0.062
	Other Malignancies	8	7.8	17	19.3	15	22.1	40	15.5	
	Non oncologic diagnosis	24	23.5	23	26.1	17	25.0	64	24.8	
Metastasis	Yes	9	8.8	16	18.2	15	22.4	40	15.6	0.042
	No	93	91.2	72	81.8	52	77.6	217	84.4	

## Data Availability

The data presented in this study are available on request from the corresponding author. The data are not publicly available due to privacy issues.

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
