# Peer review of "Lymphadenopathies before and during the Pandemic COVID-19: Increasing Incidence of Metastases from Solid Tumors"

_jcm, 2022, doi:10.3390/jcm11236979_

Round 1
Reviewer 1 Report
Overall, this was a relatively simple study which provided important data on a topic of interest to many. It adds to the growing body of literature on the topic and supports prior work detailing the effect of COVID pandemic on diagnosis and outcomes of malignancies. To my knowledge no similar published works on lymphadenopathy in general, though results are similar to publications on diagnosis of advanced breast cancer in the COVID age.
Some comments:
Introduction:
-Was not initially clear to me that the impetus for this study was that two patients with unusual distant lymph node involvement were diagnosed at the study site, thus leading to further research. I understood this more after reading discussion. May wish to expand on this more
Results:
-Some areas of wording in Results section confusing and should be revised. Took me a couple read-throughs to understand the data presented "One hundred and two patients upfront 3765 visits (2.7%)" meant that of 3765 visits during this period, 102 were for patients with lymphadenopathy who underwent excisional biopsy.
-I would clarify lines 90-91 on page 3 in Results. I am not certain what these results refer to. Six patients with distant lymph node involvement are a subset of which group?
Discussion:
-There is a noted connection between lymphadenopathy and COVID vaccination. Is it possible that any initial concerns for lymphadenopathy were attributed to COVID vaccination and therefore may have contributed to delays in seeking care/diagnosis?
-It would be interesting to know at which step in diagnosis there is a delay of care. Is this due to a delay in patient seeking initial medical care vs a delay in patient referral to Hematology vs delay in proceeding with diagnostic biopsy? That data may not all be available but if present could be important in helping to address this issue.
-In many institutions there was difficulty with scheduling procedures due to many hospitals cancelling outpatient or elective surgeries for a time. Is there any data regarding whether biopsies were offered less frequently or more difficult to obtain in this period? If excisional biopsy was more difficult to arrange during this time perhaps only more ill patients were scheduled to undergo procedures, selecting for a potentially more ill population with later stage disease
-Would use care with the term 'significant' in several areas of discussion. This term implies a statistical difference. In lines 148-150 I do not believe that the incidence of distant metastatic involvement not contiguous from primary site evaluated statistically.
-Other potential limitations of this study include single center data. Were there any possible changes in referral patterns due to travel limitations which may have affected outcomes?
Reviewer 2 Report
A retrospective study to evaluate the association between COVID-19 pandemic on the rate of metastatic lymphadenopathy (LAP). The study showed a positive association.
Comments:
1. It is stated that screening visits and surgical biopsies were not affected by the pandemic. And the rates of metastatic diseases significantly increased. I wonder how these two finding gave rise the conclusion that "the pandemic affected screening of early cancer detection." I request the authors explain it.
2. Introduction: If we consider the clinicians as the main audience of this article, the Introduction section is too long subtracting the concentration from the main endpoint of this study. It is recommended to limit the introduction of COVID-19 to maximum two sentences, and go straight to the importance of the pandemic in patients with cancer.
3. One crucial point needs to be noted. Do cancer patients with LAP had COVID-19 concomitantly or in short-term before LAP?
4. In the Methods: "Patients with
previous history of cancer or hematological malignancy were excluded".
However, in the Results:
"In this group 48 had hematological malignancies (54.5%), 17 metastatic cancers (19.3%), ...". These two information seems to in contrast. More clarification is required.
5. Figure 1B doesn't follow the information given in Table 1. For example, the rate of metastasis in 2021 is 22%; however, in Fig. 1this rate is below 20%.
6. To define the Covid pandemic as a risk factor of metastasis, it is essential to consider all the possible predictive factors of metastasis by running multivariate analysis. Unfortunately, this analysis has been missed in this study, and authors put this interesting conclusion (the association between covid and metastasis) by only a simple Chi-square test. This may not be acceptable in today's practice.
7. Discussion section needs substantial improvement. It is recommended to point out the possible mechanisms underlying the increase in LAP in the context of COVID-19. One mechanism is as follows:
- In patients with cancer, immune cells are in battle with tumor cells (https://pubmed.ncbi.nlm.nih.gov/36292613/)
- COVID-19 results in mitochondrial dysfunction and immune cells' exhaustion. (https://pubmed.ncbi.nlm.nih.gov/33510633/)
- Tumor cells evade from the exhausted immune cells (https://pubmed.ncbi.nlm.nih.gov/34890822/)
It is recommended mention this mechanism and cite the aforementioned articles.
Round 2
Reviewer 2 Report
The comments are convincingly addressed.